# Trustworthy LLM-Based Medical Decision-Making Framework: An Iterative Validation Methodology with Safety Guarantees

Makhmasoatov Elbek Shoxli ogli — e.makhmasoatov@nsu.ru

Andrey Nechesov — nechesoff@gmail.com

Artificial Intelligence Center, Novosibirsk State University, 630090 Novosibirsk, Russia
International Artificial Intelligence Committee (IAIC), Dubai, UAE
Russian Engineering Academy (IAE), Moscow, Russia

## Abstract

This article proposes a trustworthy AI framework with a task-oriented and iterative approach based on Large Language Models (LLM) for decision-support systems in the medical domain. In the proposed model, the initial medical problem is first analyzed using an LLM, based on which multiple potential solutions $t_1, t_2, \ldots, t_k$ are generated with transparent reasoning traces. These solutions are then rigorously validated according to their correctness, that is, their ability to provide a logically complete, well-justified, and verifiable solution to the problem. If a solution passes validation, the system generates the final output with full provenance tracking and terminates. Otherwise, a safety-critical constraint condition $N$ determines whether to continue, considering iteration limits, computational resources, and safety boundaries to prevent harmful outputs. If $N$ is satisfied, the problem is reformulated and iterative analysis continues; otherwise, the system fails safely without output. This trustworthy AI approach reduces the risk of incorrect or premature conclusions while enhancing reliability, explainability, verifiability, and safety of LLM-based medical decision-support systems.

Keywords: Large Language Models; Trustworthy AI; Medical Decision Support; Iterative Validation; Explainable AI; Safety Constraints; Clinical Diagnosis; Solution Verification; Healthcare AI; Reliable Machine Learning

## 1 Introduction

Artificial Intelligence (AI) has evolved significantly since the Dartmouth workshop, where the field was formally introduced by McCarthy

The integration of LMS into medical decision support systems creates a unique set of problems that go beyond the usual limitations of artificial intelligence. In medicine, the consequences of inaccurate or misleading results can be fatal, misdiagnosed, improper treatment, and harmful to patients. Making a medical decision requires not only predictive accuracy, but also logical consistency, verifiability, interpretation, safety assurances, and managed failure mechanisms. A system that cannot explain its reasoning or verify its conclusions contrary to existing medical knowledge cannot be trusted in clinical conditions.

Currently, one of the main problems of medical systems based on LLM is their inability to resolve logical contradictions. When presented with conflicting symptoms with some diagnoses, these models often fail to recognize and resolve these inconsistencies. For example, a patient who came with symptoms characteristic of pregnancy but identified as male should not be diagnosed with pregnancy, but LLM can still lead to such conflicting outcomes due to its statistical nature. This problem stems from the fact that LLMs operate on the basis of probabilistic correlations, not formal logical reasoning.

Another important issue is the lack of verification at LLM outputs. Traditional medical decision-making relies on traditional clinical guidelines and evidence-based practice. When a doctor makes a diagnosis, they may provide medical knowledge that supports certain symptoms, test results, and their conclusion. In contrast, LLM-based systems function as potential black boxes

The lack of security guarantees in current systems is probably the most important barrier to clinical acceptance. When faced with uncertain or insufficient information, human doctors have the opportunity to acknowledge the uncertainty and order additional tests or consultations. However, systems based on LLM are usually designed to diagnose medesena always, even if the available evidence is insufficient. Forced exit can lead to reliably presented but inaccurate diagnoses, creating a dangerous situation where users may unreasonably trust system recommendations.

Furthermore, existing systems lack structured validation mechanisms that could catch errors before they reach the user. While a physician might mentally check their diagnosis against known contraindications and diagnostic criteria, LLMs typically generate a single response without any internal verification step. This one-shot approach means that if the initial generation contains errors, there is no opportunity to recognize and correct them through systematic checking against medical knowledge bases.

The problem is complicated by the fact that medical knowledge is inherently hierarchical and interconnected. A simple statistical correspondence between symptoms and diagnoses cannot encompass complex relationships that characterize clinical considerations. For example, the presence of some symptoms, regardless of how well the other symptoms match, may preclude definite diagnoses, and in many cases, even when individual symptoms appear, other combinations of symptoms may be required to confirm the diagnosis. Such a logical structure of medical knowledge cannot be sufficiently expressed in statistical approaches or applied in its pure form.

Trustworthy AI principles, as defined by NIST

In this work, we address these issues by proposing a task-oriented, iterative, and security-dependent structure for making medical decisions based on LLM. Our approach introduces a number of key innovations designed to overcome the limitations of current systems. Firstly, we implement a multi-iteration architecture that allows the system to improve its output based on the accumulated knowledge and logical constraints. Rather than one attempt, our system can make N attempts each time to improve the results, including data from previous iterations.

Second, we integrate explicit logical validation mechanisms that check generated diagnoses against formal medical knowledge bases. Using two types of rules - negative rules that identify contradictions and positive rules that confirm diagnostic criteria - our system ensures that every suggested diagnosis is logically consistent with established medical knowledge. This validation occurs after each generation attempt, providing a structured verification layer that catches errors before they reach the user.

Thirdly, we will introduce a security shutdown mechanism that will allow the system to acknowledge its failure to reach a reliable conclusion. If a diagnosis is not made after N iterations and it fails both negative and positive rule validation, the system will output 'No reliable diagnosis found' instead of forcing an inaccurate or potentially harmful result. This controlled failure mechanism is crucial and necessary for ensuring safety in clinical guidelines.

Fourth, our framework maintains complete transparency by tracking all rules and diagnoses used in each iteration. This provenance information enables users to examine the system's reasoning process, understand why certain diagnoses were rejected or accepted, and verify the logical basis for final recommendations. Such transparency is crucial for building trust in AI-assisted medical decision-making.

Our framework extends the logical-probabilistic learning approach and knowledge hierarchy model introduced in IEEE Learning Theory

## 2 Related Work

The application of Large Language Models (LLMs) in medicine has developed rapidly in recent years. The transformer architecture proposed by Vaswani et al.

Vityaev et al.

Goncharov and Nechesov

Recent research is exploring verification mechanisms for clinical AI. Retrieval-Augmented Generation (RAG) approaches attempt to reduce hallucinations by verifying AI outputs against trusted medical databases. However, this verification typically occurs once, after generation, rather than during the reasoning process itself.

Research on iterative consultation between two independent AI systems has shown that initial disagreements can be reduced by up to 10% after several iterations. This suggests that iterative approaches can achieve autonomous error detection and increase reliability.

More specifically, we propose a framework with five iterations:

Multi-iterative Architecture: Our system can generate diagnoses up to five times. Each iteration builds upon the knowledge accumulated from previous iterations. This allows for the gradual refinement of results.

Two Types of Logical Rules: We utilize negation rules (which identify contradictions and reject diagnoses) and confirmation rules (which confirm diagnoses when sufficient evidence exists). These rules are derived from formal medical knowledge bases.

Safe Stopping Mechanism: If, after five iterations, a diagnosis still does not pass the verification of both negation and confirmation rules, the system stops and outputs "No reliable diagnosis found," rather than forcing an uncertain conclusion.

Inter-iteration Knowledge Transfer: All rules and diagnoses used in each iteration are collected and passed on to subsequent iterations. This enables the system to learn from its own verification history.

Integration of Logical and Probabilistic Reasoning: Based on Goncharov and Nechesov

Thus, our system combines the pattern recognition capabilities of LLMs with the logical rigor of formal medical knowledge and provides safety guarantees through a controlled stopping mechanism. Unlike purely neural systems, our approach ensures that every output is verified against explicit rules before being generated, addressing a key gap in existing research.

## 3  Mathematical Formulation of the Trustworthy Iterative Framework

### 3.1  Medical Task Representation

Let the medical decision problem be formalized as
$$\forall x \, \exists y \, F(x, y), \tag{1}$$
where $x$ is the patient symptom set, $y$ is the diagnosis, and $F(x, y)$ is the diagnostic validity condition.

We define
$$F(x, y) : \Phi(x, y) \to \Psi(x, y), \tag{2}$$
where $\Phi(x, y)$ is observed symptom consistency and $\Psi(x, y)$ is diagnostic conclusion validity.

### 3.2  Basic Notation

Set of symptoms:
$$S = \{s_1, s_2, s_3, \ldots, s_k\} \tag{3}$$
where:
$$s_i - i\text{-th symptom}$$
$$k - \text{total number of symptoms}$$

### 3.3  Basic Notation for Diagnoses

Set of diagnoses:
$$D = \{d_1, d_2, d_3, \ldots, d_M\} \tag{4}$$
where:
$$d_j - j\text{-th diagnosis (e.g., } d_1 - \text{influenza, } d_2 - \text{COVID-19)}$$
$$M - \text{total number of possible diagnoses}$$

Diagnoses generated by LLM at iteration $t$:
$$D^{(t)} = \{d_1^{(t)}, d_2^{(t)}, d_3^{(t)}, \ldots, d_{n_t}^{(t)}\} \tag{5}$$
where:
$$t - \text{iteration number}$$
$$n_t - \text{number of diagnoses generated at iteration } t$$

Threshold value:
$$\Theta \in [0, 1] \tag{6}$$
In medical practice, $\Theta = 0.7$ is typically accepted. This means that only rules and diagnoses with 70% or higher confidence level are accepted.

Maximum number of iterations:
$$N = 5 \tag{7}$$
The iterative process is controlled by the condition $I < N$, where:
$$I - \text{current iteration number}$$
$$N - \text{maximum allowed number of iterations}$$

### 3.4 Logical Rule System

Our system employs two types of logical rules to ensure trustworthy and verifiable medical decision-making. These rules are derived from formal medical knowledge bases and serve to validate and confirm diagnoses generated by the LLM.

#### 3.4.1 Negation Rules (Exclusion Rules)

Negation rules indicate that a specific set of symptoms excludes a particular diagnosis. In other words, if a patient exhibits all the symptoms specified in the rule, then the corresponding diagnosis cannot be made.

Mathematical Representation:

$$R_\neg = \{r_\neg^1, r_\neg^2, r_\neg^3, \ldots, r_\neg^m\} \tag{8}$$

Each negation rule has the following form:

$$r_\neg^i : (s_a \wedge s_b \wedge s_c \wedge \ldots \to \neg d_x, p_i) \tag{9}$$

where:

- $s_a, s_b, s_c, \ldots$ - symptoms (e.g., "male", "pregnancy test positive")
- $\neg d_x$ - not diagnosis $d_x$ (exclusion of this diagnosis)
- $p_i$ - confidence level of the rule (from 0 to 1)

#### 3.4.2 Confirmation Rules

Confirmation rules indicate that a specific set of symptoms confirms a particular diagnosis. If a patient exhibits all the symptoms specified in the rule, then the corresponding diagnosis can be made.

Mathematical Representation:

$$R_+ = \{r_+^1, r_+^2, r_+^3, \ldots, r_+^l\} \tag{10}$$

Each confirmation rule has the following form:

$$r_+^i : (s_a \wedge s_b \wedge s_c \wedge \ldots \to d_y, p_i) \tag{11}$$

where:

- $s_a, s_b, s_c, \ldots$ - symptoms (e.g., "fever", "cough", "headache")
- $d_y$ - diagnosis $d_y$ (e.g., "influenza")
- $p_i$ - confidence level of the rule (from 0 to 1)

### 3.5 Significance of the Rule System

These two types of logical rules constitute the core validation mechanism of our system:

Negation Rules:

- Identify and prevent logical contradictions
- Eliminate mutually exclusive diagnoses from being simultaneously considered
- Filter out medically impossible cases

Confirmation Rules:

- Verify the validity of diagnoses
- Accept only diagnoses supported by sufficient evidence
- Ensure statistical reliability

In our approach, every diagnosis generated by the LLM undergoes a two-stage validation process: first against negation rules, then against confirmation rules. This dual-stage verification system addresses one of the most significant limitations of LLMs—their inability to detect logical contradictions. Furthermore, it guarantees the validity and reliability of medical decisions by ensuring that every output is grounded in formal medical knowledge and supported by adequate evidence.

### 3.6 Theoretical Foundation of the Iterative Process

### 3.7 Basic Definitions and Notation

We introduce the following notation:

- $S = \{s_1, s_2, \ldots, s_n\}$ - set of patient symptoms, $|S| = n$
- $KB = (R_\neg, R_+)$ - medical knowledge base, $|KB| = m$, where:
  - $R_\neg = \{r_\neg^1, \ldots, r_\neg^{m_1}\}$ - negation rules (contraindications)
  - $R_+ = \{r_+^1, \ldots, r_+^{m_2}\}$ - confirmation rules (diagnostic criteria)
  - $m_1 + m_2 = m$
- $D^{(t)}$ - set of diagnoses generated by LLM at iteration $t$, $|D^{(t)}| \leq k$, $k = const$
- $V^{(t)} \subseteq D^{(t)}$ - set of verified diagnoses at iteration $t$
- $N \in \mathbb{N}$ - maximum number of iterations (typically $N = 5$)
- $\Theta \in [0, 1]$ - confidence threshold (typically $\Theta = 0.7$)

### 3.8 Theorem (Computational Complexity of Iterative Diagnosis)

Theorem: For a given set of symptoms $S$ and knowledge base $KB$, the diagnostic process with $N$ iterations:

(i) Terminates after at most $N$ iterations;

(ii) Has computational complexity $O(n \cdot m^{p+1})$, where $p \in \mathbb{N}$ is a constant;

(iii) Returns either a verified diagnosis $d^* = \arg\max_{d \in V^{(t)}} p(d)$ when $V^{(t)} \neq \emptyset$, or the message "No reliable diagnosis found" when $V^{(t)} = \emptyset$ for all $N$ iterations.

Proof.

1. Termination. The process runs from $t = 1$ to $t = N$. After each iteration, the value of $t$ is incremented by one. When $t = N$ is reached, the for loop terminates and the process stops. Thus, termination is guaranteed.

2. Complexity of LLM generation. According to the result of Goncharov and Nechesov [43], a single run of a fully connected neural network or LLM takes $O(n \cdot k)$ time, where $k$ is the number of generated diagnoses (in practice, $k \leq 10$). Since $k$ is constant, this can be written as $O(n)$.

3. Complexity of logical verification. Let each rule be of the form $r : \Phi(x, y) \to \Psi(x, y)$. Checking whether the condition $\Phi(x, y)$ is satisfied requires $O(n \cdot |\Phi|^p)$ operations, where $p$ is the complexity degree of the rule (typically $p \leq 3$). Since $|\Phi| \leq m$, we can write $O(n \cdot m^p)$.

For each diagnosis:

- Checking all $R_\neg$ rules: $O(m_1 \cdot n \cdot m^p) = O(n \cdot m^{p+1})$
- Checking all $R_+$ rules: $O(m_2 \cdot n \cdot m^p) = O(n \cdot m^{p+1})$

Since $m_1 + m_2 = m$, the total is $O(n \cdot m^{p+1})$.

At each iteration, $k$ diagnoses are verified:

$$O(k \cdot n \cdot m^{p+1}) = O(n \cdot m^{p+1}), \quad \text{because } k = const.$$

4. Number of iterations. The process consists of $N$ iterations. $N = 5$ is constant. Therefore, the total complexity is:

$$O(N \cdot n \cdot m^{p+1}) = O(n \cdot m^{p+1}).$$

5. Correctness of the result. If $V^{(t)} \neq \emptyset$, then every diagnosis in this set:

- Is not rejected by any negation rule $R_\neg$
- Is confirmed by at least one confirmation rule $R_+$

Thus, they are consistent with the medical knowledge base. Among them, the most reliable one $d^* = \arg\max p(d)$ is selected.

If $V^{(t)} = \emptyset$ for all $N$ iterations, this means that for the given symptoms, no diagnosis consistent with the knowledge base was found. In this case, outputting the message "No reliable diagnosis found" ensures safe termination.

Thus, the theorem is completely proven. ∎

## 4 System Architecture and Implementation

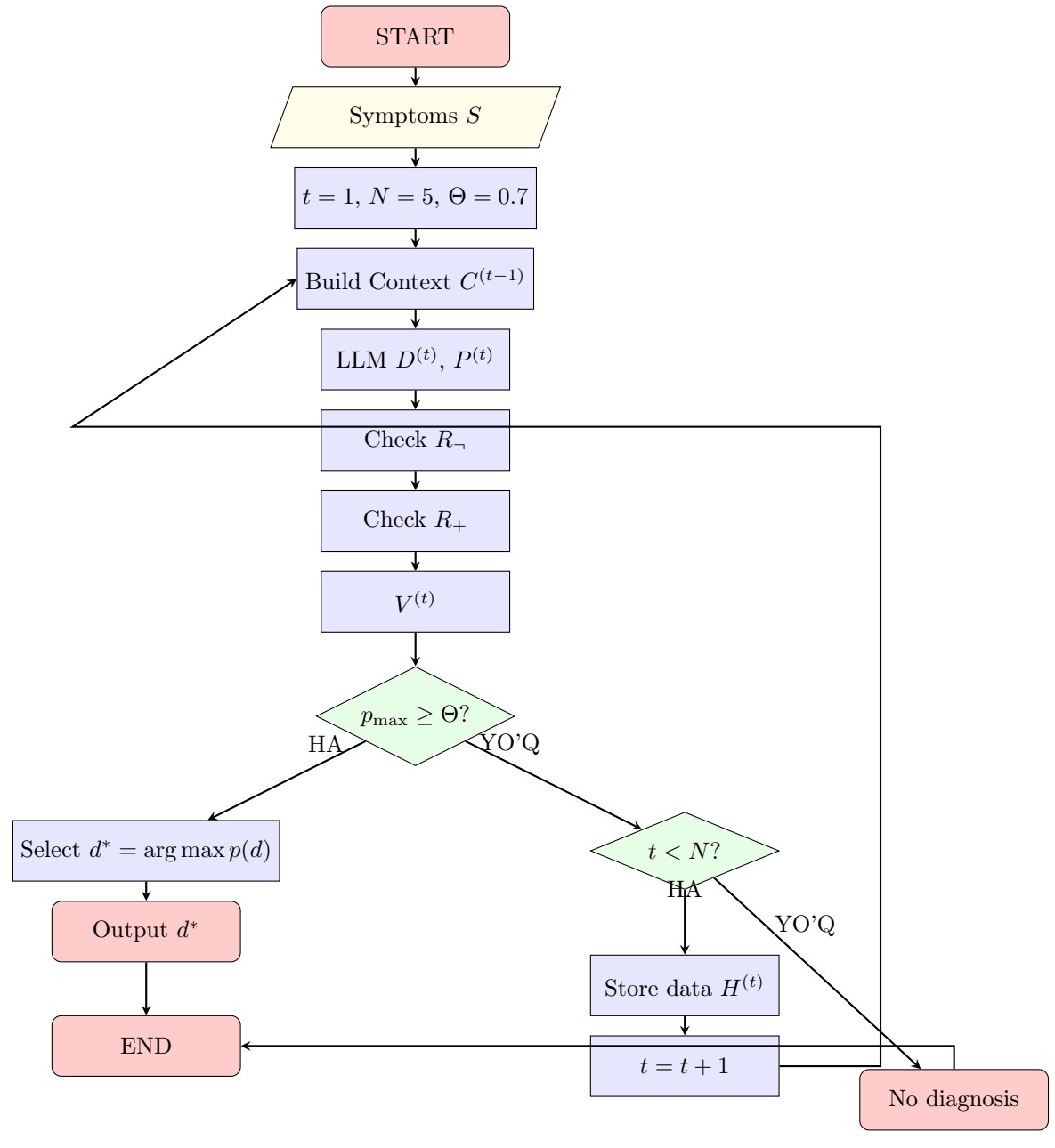

Figure 1: Iterative LLM diagnosis algorithm with START and END

> Algorithm explanation:
> • START begins, symptoms $S$ are input
> • YES branch: $p_{\max} \geq 0.7 \rightarrow$ Diagnosis found and END
> • NO $\rightarrow t < N?$ YES: $t = 1, 2, 3, 4 \rightarrow$ New iteration (red line)
> • NO $\rightarrow t < N?$ NO: $t = 5 \rightarrow$ "No diagnosis" and END

Our framework implements a five-iteration diagnostic process and combines Large Language Models with precise logical verification mechanisms. The system consists of four main components: Medical

Knowledge Base, LLM Interface, Verification Mechanism, and Iterative Controller. This section presents the core implementation of LLM integration and iterative control mechanisms.

The LLM interface manages communication with the language model and constructs context-based queries that incorporate information from previous iterations. At each iteration $t$, the LLM receives the patient's symptoms $S_p$, threshold value $\Theta$, and context $C^{(t-1)}$ accumulated from previous attempts.

The key innovation is iterative context building: each LLM call receives information about which diagnoses have been considered and which rules have been applied in previous iterations, enabling progressive improvement of results across attempts.

### 4.1 Core Algorithm: Five-Iteration Diagnostic Process

The complete diagnostic process is presented in Algorithm 1, which formalizes the iterative refinement logic:

---

**Algorithm 1** Iterative LLM-based Diagnosis and Verification

---

**Require:** Patient symptoms $S_p$, threshold value $\Theta$, maximum iterations $N = 5$
**Ensure:** Diagnosis $d^*$ or safe stopping
 1: **for** $t = 1$ to $N$ **do**
 2:    // Build context from previous iterations
 3:    $C = \{\text{previous diagnoses, applied rules}\}$
 4:    // Generate candidates using LLM
 5:    $D^{(t)} = \text{LLM}(S_p, \Theta, C)$
 6:    // Verify each candidate
 7:    $V^{(t)} = \{\}$
 8:    **for** each $d$ in $D^{(t)}$ **do**
 9:      **if** no negation rule rejects $d$ **then**
10:        **if** affirmation rule supports $d$ **then**
11:          $V^{(t)} = V^{(t)} \cup \{d\}$
12:        **end if**
13:      **end if**
14:    **end for**
15:    // Return if diagnoses are found
16:    **if** $V^{(t)} \neq \emptyset$ **then**
17:
18:      **return** $\arg\max_{d \in V^{(t)}} p(d)$
19:    **end if**
20: **end for**
21:
22: **return** "No confident diagnosis found" // Safe stopping

---

If no diagnosis is found after five iterations, the system safely halts. This prevents the generation of uncertain and potentially dangerous conclusions.

## 5 Conclusion

In this paper, we have proposed a trustworthy Artificial Intelligence (AI) framework for medical decision-support systems. This framework integrates Large Language Models (LLMs) with formal logical verification mechanisms. Our approach addresses the fundamental limitations of current LLM-based systems, while also overcoming problems such as lack of logical transparency, unreliable reasoning, and absence of safety guarantees.

The N-repetitive diagnostic architecture allows for step-by-step clarification of diagnoses through accumulated contextual knowledge. This allows the system to learn from its own verification history. A two-step logical confirmation mechanism, consisting of rule of negation that identifies contradictions and rule of confirmation that checks diagnostic criteria, ensures that each output logically corresponds to established medical knowledge. The Safe Stopping Protocol prevents the emergence of fuzzy or potentially harmful conclusions if no diagnosis is confirmed even after N iterations.

The system architecture consists of four main components: Medical Knowledge Base, LLM Interface, Verification Mechanism, and Iterative Controller. These components implement a transparent and verifiable diagnostic process. By tracking all rules and diagnoses used in each iteration, the framework provides complete provenance information, enabling users to examine the system's reasoning process and verify the logical basis for final recommendations.

The main contributions of this work are:

- A novel multi-iteration architecture that enables progressive refinement of diagnoses through inter-iteration knowledge transfer
- A dual-stage logical verification system using negation and affirmation rules derived from formal medical knowledge bases
- A safety-constrained stopping mechanism that prevents forced output generation when a reliable conclusion cannot be reached
- Ensuring transparency through complete tracking of all rules and diagnoses across iterations
- Integration of logical-probabilistic reasoning with statistical inference for robust medical decision-making

Future work will focus on several directions: extending the framework to handle complex medical scenarios with interacting comorbidities, incorporating temporal dynamics of symptom progression, validating the approach on real clinical datasets, and developing user interfaces to effectively communicate the system's reasoning process to medical professionals. Additionally, we plan to integrate uncertainty quantification methods to provide confidence intervals for diagnostic recommendations and explore the framework's applicability to other safety-critical domains beyond healthcare.

In conclusion, our trustworthy AI framework demonstrates that it is possible to harness the power of Large Language Models for medical decision support while maintaining the logical consistency, verifiability, and safety guarantees that clinical applications demand. By explicitly modeling the iterative nature of diagnostic reasoning and enforcing logical constraints through formal verification, we provide a path toward AI systems that can truly assist physicians in making accurate, reliable, and safe medical decisions.

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
