# OpenReview forum: "Trustworthy LLM-Based Medical Decision-Making Framework: An Iterative Validation Methodology with Safety Guarantees"
_mathai.club/MathAI/2026/Conference — 2026 Oral_

### Official Review · Reviewer_b7H7 · 2026-03-11
**Novel iterative framework for trustworthy LLM diagnosis — good ideas, but no experiments to back them up**

**Rating:** 5
**Confidence:** 4

**Review:**

This paper tackles an important problem — how to make LLM-based medical diagnosis more reliable and safe. Applying LLMs in clinical practice comes with well-known issues: hallucinations, lack of logical consistency, and weak safety mechanisms. The authors propose a way to address these problems.

The introduction does a good job laying out these challenges — from the "black box" problem to the dangers of forced outputs when evidence is insufficient. It clearly motivates why we need logical verification and safety mechanisms in medical AI.

Their idea is an iterative approach: the system generates diagnoses over several rounds, each time checking them against logical rules — negation rules to filter out contradictions, and confirmation rules to ensure enough evidence is present. If after five iterations no diagnosis passes the checks, the system outputs "no reliable diagnosis found" instead of guessing. They also keep track of all steps, so a doctor could follow the reasoning.

While iterative prompting and RAG (Retrieval-Augmented Generation) are not new by themselves, the contribution here lies in the combination of several elements:

Multi-iteration architecture with knowledge transfer — the system accumulates context across iterations, learning from its own verification history.

Dual-stage logical rule system — explicitly separating negation rules (to exclude contradictions) and confirmation rules (to affirm diagnoses) is a novel and well-motivated design choice.

Safe stopping mechanism — allowing the system to abstain from diagnosis when confidence is low is a critical safety feature for clinical practice.

Complete provenance tracking — transparency through traceability addresses the "black box" criticism of neural models.

The multi-iteration approach with knowledge transfer is a good solution, and the safe stopping mechanism is exactly what you'd want in a clinical setting. The mathematical formalization in Section 3 deserves respect (e.g., the complexity theorem).

The serious drawback of this paper is the complete absence of experimental validation.

The paper describes a system, proves a complexity theorem, and claims it "addresses key gaps in existing research" — but provides no empirical evidence that it actually works. Key questions remain unanswered:

On what datasets (synthetic or real) was the system tested?

Which specific LLM was used (GPT-4, LLaMA, something else)?

What are the quantitative results (accuracy, precision, recall, F1, reduction in hallucination rate)?

How does the iterative approach compare to single-shot LLM diagnosis or RAG-based methods?

What are the computational costs of running five iterations in practice?

The authors themselves acknowledge this in the "Future Work" section ("validating the approach on real clinical datasets") — but a paper submitted for publication should include at least preliminary experimental results to support its claims. As it stands, this is an interesting and well-argued hypothesis, not a validated solution.

Pros:

Relevant problem

Original architecture (multi-iteration + knowledge transfer)

Dual rule system

Safe stopping — a critical feature

Transparency through provenance tracking

Solid theoretical foundation

Well-written introduction that clearly motivates the work

Cons:

No experimental validation

Missing details on the knowledge base source and rule extraction

Unsupported claim about the confidence threshold

Unclear how exactly the LLM is integrated (fine-tuning or prompting?)

Shallow literature review

No critical discussion of limitations

Recommendation:

Needs revision. The ideas are clear, the theoretical part is solid, but if you're proposing a new system, it would be nice to show that it actually works. Without experiments, it's just a hypothesis.

---

### Official Review · Reviewer_sqNo · 2026-03-12
**Good evidence base, experimental is required**

**Rating:** 6
**Confidence:** 4

**Review:**

Strengths:
Section 3 provides a rigorous mathematical formulation of the diagnostic process, including set designations, rule definitions, and the complexity theorem. This adds to the theoretical validity of the concept.
The inclusion of a controlled failure output signal ("No reliable diagnosis has been established") is a reliable safety feature that reduces the risk of harm caused by false alarms.

Weaknesses:
The framework's success hinges entirely on the existence of a comprehensive, formalized medical knowledge base containing explicit negation and confirmation rules.

This paper presents a promising and well-founded system for improving the reliability of LLM-based medical diagnostics. Its strengths lie in multi-stage development, logical verification, and security mechanisms. However, in order for it to be published in a reputable publication, empirical verification, clearer implementation details, and a deeper study of existing literature are required. With these improvements, the work can make a significant contribution to the creation of reliable clinical artificial intelligence.

Questions: Have you implemented any prototype or conducted preliminary experiments? If not, how can you claim that the framework improves reliability without empirical evidence?

---

### Official Review · Reviewer_q5jC · 2026-03-13
**Marginally above acceptance threshold.  Trustworthy LLM-Based Medical Decision-Making Framework: An Iterative Validation Methodology with Safety Guarantees**

**Rating:** 6
**Confidence:** 4

**Review:**

This manuscript is devoted to an important problem of reducing the risk of incorrect or premature conclusions while enhancing reliability, explainability, verifiability, and safety of LLM-based medical decision-support systems. The author proposes an original trustworthy AI framework with a task-oriented and iterative approach based on Large Language Models (LLM) for decision-support systems in the medical domain. Making a medical decision requires not only predictive accuracy, but also logical consistency, verifiability, interpretation, safety assurances, and managed failure mechanisms. The proposed AI framework, including a few operation steps of analysis, seems to be useful since it takes into accounts these requirements.
1.	Mathematical Rigor: high.
2.	Novelty & Contribution: good.
3.	Relevance to MathAI: high.
4.	Technical Quality: good.
5.	Clarity & Presentation: good.
6.	AI-Generation Risk: very low

---

### Decision · Program_Chairs · 2026-03-14

**Decision:**

Accept (Oral)

**Comment:**

Dear Author(s),

On behalf of the Program Committee of the International Conference on Mathematics of Artificial Intelligence (MathAI 2026), we are pleased to inform you that your paper has been accepted for an oral presentation at MathAI 2026.

Your paper was evaluated through a rigorous two-stage review process involving both automated screening and expert review by members of the Program Committee. The reviewers recognized the quality and contribution of your work.

Presentation details:

- Format: Oral presentation (15–20 minutes + 5 minutes Q&A)
- Mode: You may present either in person (offline) at the conference venue in Sirius, Russia, or remotely via Zoom. Please indicate your preferred mode when confirming your participation.
- Conference dates: Marh 30 - April 3, 2026
- Website: https://mathai.club

Next steps:

1. Please confirm your participation and presentation mode by replying to this email mathai.club@yandex.ru no later than March 15, 2026 18:00 Moscow time.
2. If you plan to attend in person, the organizing committee will provide accommodation details separately.
3. Please prepare your final camera-ready manuscript according to the formatting guidelines available at https://mathai.club and upload it to OpenReview by March 15, 2026 18:00 Moscow time.

Should you have any questions regarding the program, logistics, or your presentation slot, please do not hesitate to contact us.

We look forward to your contribution to MathAI 2026.

With kind regards,

MathAI 2026 Program Committee
International Conference on Mathematics of Artificial Intelligence
https://mathai.club
OpenReview: https://openreview.net/group?id=mathai.club/MathAI/2026/Conference
Telegram: https://t.me/MathAI_club
Email: mathai.club@yandex.ru